# Circadian regulation of sinoatrial nodal cell pacemaking function: Dissecting the roles of autonomic control, body temperature, and local circadian rhythmicity

Pan Li[1]*, Jae Kyoung Kim[1,2]*

1 Biomedical Mathematics Group, Pioneer Research Center for Mathematical and Computational Sciences, Institute for Basic Science, Daejeon, Republic of Korea, 2 Department of Mathematical Sciences, KAIST, Daejeon, Republic of Korea

* panli@ibs.re.kr (PL); jaekkim@kaist.ac.kr (JKK)

**Data Availability Statement:** Model codes are provided for public download from https://github.com/Mathbiomed/CircSANC.

## Abstract

Strong circadian (~24h) rhythms in heart rate (HR) are critical for flexible regulation of cardiac pacemaking function throughout the day. While this circadian flexibility in HR is sustained in diverse conditions, it declines with age, accompanied by reduced maximal HR performance. The intricate regulation of circadian HR involves the orchestration of the autonomic nervous system (ANS), circadian rhythms of body temperature (CRBT), and local circadian rhythmicity (LCR), which has not been fully understood. Here, we developed a mathematical model describing ANS, CRBT, and LCR in sinoatrial nodal cells (SANC) that accurately captures distinct circadian patterns in adult and aged mice. Our model underscores how the alliance among ANS, CRBT, and LCR achieves circadian flexibility to cover a wide range of firing rates in SANC, performance to achieve maximal firing rates, while preserving robustness to generate rhythmic firing patterns irrespective of external conditions. Specifically, while ANS dominates in promoting SANC flexibility and performance, CRBT and LCR act as primary and secondary boosters, respectively, to further enhance SANC flexibility and performance. Disruption of this alliance with age results in impaired SANC flexibility and performance, but not robustness. This unexpected outcome is primarily attributed to the age-related reduction in parasympathetic activities, which maintains SANC robustness while compromising flexibility. Our work sheds light on the critical alliance of ANS, CRBT, and LCR in regulating time-of-day cardiac pacemaking function and dysfunction, offering insights into novel therapeutic targets for the prevention and treatment of cardiac arrhythmias.

## Author summary

The mammalian heart relies on the sinoatrial node, known as the cardiac pacemaker, to orchestrate heartbeats. These heartbeats slow down during sleep and accelerate upon waking, in anticipation of daily environmental changes. The heart's ability to rhythmically

**Funding:** This work was supported by Institute for Basic Science IBS-R029-C3 (to J.K.K.). The funder had no role in study design, data collection and analysis, decision to publish, or preparation of the manuscript.

**Competing interests:** The authors have declared that no competing interests exist.

adapt to these 24-hour changes, known as circadian rhythms, is crucial for flexible cardiac performance throughout the day, accommodating various physiological states. However, with aging, the heart's circadian flexibility gradually weakens, accompanied by a decline in maximal heart rate. Previous studies have implicated the involvement of a master circadian clock and a local circadian clock within the heart, but their time-of-day interactions and altered dynamics during aging remain unclear. In this study, we developed a mathematical model to simulate the regulation of sinoatrial nodal cell pacemaking function by the master and local circadian clocks in adult and aged mice. Our results unveiled distinct roles played by these clocks in determining circadian patterns of sinoatrial nodal cells, shedding light on their critical alliance in regulating time-of-day cardiac pacemaking function and dysfunction.

## Introduction

The mammalian heart exhibits robust circadian rhythms in various cardiac function indices, such as heart rate (HR) and electrocardiogram waveforms [1]. During sleep, HR slows down, accompanied by prolongation of QRS duration and QT interval, while it accelerates upon waking, indicating circadian variations in the electrical properties of cardiac function [2]. In addition, cardiac arrhythmic phenotypes also show distinct circadian patterns. For instance, while bradyarrhythmias and Brugada syndrome are more prevalent at night, ventricular fibrillation and sudden cardiac death are more common in the morning [2,3].

These circadian rhythms (~24hr) in cardiac physiology and pathology are regulated by the master circadian clock located in the suprachiasmatic nucleus (SCN) [3]. The master clock in the SCN modulates time-of-day heartbeats by influencing the firing rate (FR) of the sinoatrial node, known as the cardiac pacemaker, through the autonomic nervous system (ANS) [4]. At cellular level, each heartbeat (~ hundreds of milliseconds) is initiated by an action potential (AP) generated in a sinoatrial nodal cell (SANC). The FR of SANC AP is intricately regulated through the delicate interplay between a membrane oscillator (MO) and a $Ca^{2+}$ oscillator (CO) (Fig 1). The MO is associated with sarcolemmal ionic channels, exchangers and pumps, while the CO involves intracellular $Ca^{2+}$ release from the sarco/endoplasmic reticulum (SR) into the cytosol, and $Ca^{2+}$ removal through SR $Ca^{2+}$ ATPase (SERCA) (Fig 1). Both MO and CO of SANC are tightly regulated by the ANS, influencing the circadian variation in SANC FR (Fig 1). Specifically, sympathetic nervous activities (SNA) increase SANC FR by enhancing both MO and CO through the activation of cyclic adenosine monophosphate (cAMP)—protein kinase A (PKA) signaling pathway. This cAMP-PKA signaling pathway leads to the phosphorylation of various MO targets, such as the L-type $Ca^{2+}$ channel ($I_{CaL}$), T-type $Ca^{2+}$ channel ($I_{CaT}$), and CO targets, including Ryanodine receptor, and phospholamban that regulates the activity of SERCA (Fig 1; yellow dot) [5]. In contrast, parasympathetic nervous activities (PNA) reduce SANC FR by weakening MO through the activation of the muscarinic $K^+$ current ($I_{KACh}$), and by inhibiting the cAMP-PKA signaling pathway (Fig 1; blue dot) [6]. Both SNA and PNA display circadian variations in regulating SANC FR. Specifically, SNA peaks after awakening to increase SANC FR, whereas PNA reaches its peak during sleep to reduce SAN FR. As a result, the properties of MO and CO in SANC are delicately adjusted to generate circadian variations in SANC FR, anticipating daily environmental changes.

Besides the ANS, circadian rhythms in HR are tuned by the ups and downs of core body temperature (*BT*), which is regulated by the master clock in the SCN as well (Fig 1) [7,8]. *BT*

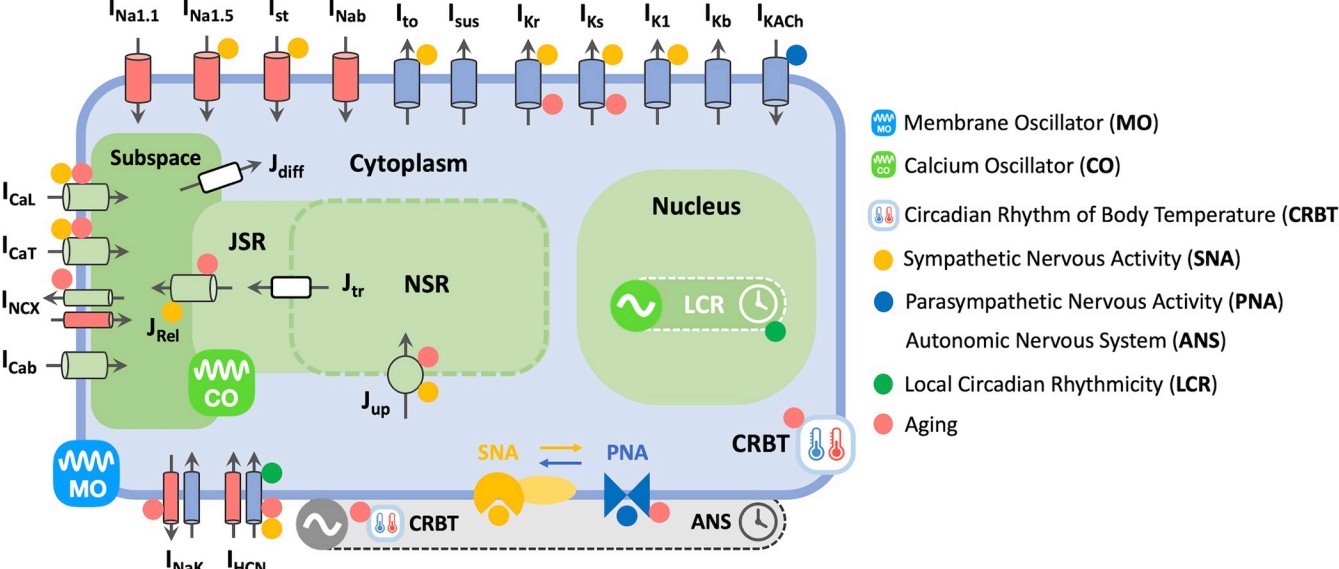

**Fig 1. Model schematic for ANS, CRBT and LCR in mouse SANC.** During each heartbeat, SANC FR is determined by coupled interactions between the MO linked to sarcolemmal ionic currents, and the CO associated with SR Ca$^{2+}$ release ($J_{Rel}$) and uptake ($J_{up}$). Over the course of a day and night cycle, the properties of MO and CO in SANC are tightly regulated by the ANS, CRBT, and LCR to generate circadian variations in FR. The ANS is regulated by the master circadian clock—SCN. Simultaneous SNA (orange dots) and PNA (blue dots) co-modulate a diverse range of subcellular targets in SANC and exhibit non-additive effects via cAMP-PKA dependent or independent pathways [6]. CRBT regulates the autonomic balance between SNA and PNA, influencing the kinetics and/or conductance of ion channels, exchangers, and pumps. LCR (green dots) within the SANC nucleus leads to circadian variations in the expression levels of ion channels, e.g., $I_{HCN}$. In aged mice, SANC pacemaking function is disrupted by aging-dependent ion channel remodeling, intracellular Ca$^{2+}$ cycling alternation, PNA impairment, and CRBT disruption (red dots) [25,26,39–41]. $I_{Na1.1}$, Na$^+$ channel isoform Na$_v$1.1 current; $I_{Na1.5}$, Na$^+$ channel isoform Na$_v$1.5 current; $I_{st}$, sustained inward Na$^+$ current; $I_{Nab}$, background Na$^+$ current; $I_{to}$, transient component of 4-Aminopyridine-sensitive current; $I_{sus}$, sustained component of 4-Aminopyridine-sensitive current; $I_{Kr}$, rapid delayed rectifying K$^+$ current; $I_{Ks}$, slow delayed rectifying K$^+$ current; $I_{K1}$, inward rectifier K$^+$ current; $I_{Kb}$, background K$^+$ current; $I_{KACh}$, muscarinic K$^+$ current; $I_{CaL}$, L-type Ca$^{2+}$ channel current; $I_{CaT}$, T-type Ca$^{2+}$ channel current; $I_{NCX}$, Na$^+$/Ca$^{2+}$ exchanger current; $I_{Cab}$, background Ca$^{2+}$ current; $I_{NaK}$, Na$^+$/K$^+$ pump current; $I_{HCN}$, hyperpolarization-activated cyclic nucleotide–gated channel current; JSR, junctional sarcoplasmic reticulum; NSR, network sarcoplasmic reticulum; $J_{diff}$, Ca$^{2+}$ diffusion flux from subspace to cytoplasm compartment; $J_{Rel}$, Ca$^{2+}$ release from JSR to subspace compartment; $J_{tr}$, Ca$^{2+}$ transfer flux from NSR to JSR; $J_{up}$, SERCA Ca$^{2+}$ pump flux. The CRBT icon is derived from https://openclipart.org/detail/231080/thermometer, while the circadian clock icon is modified based on https://openclipart.org/detail/198766/mono-tool-timer.

modulates HR by inducing temperature-dependent changes in autonomic balance, intracellular ionic diffusion processes, and the properties of ion channels, exchangers, and pumps [9–14]. For instance, experimental studies demonstrated that as BT increases, PNA declines with an increase in SNA [14]. Furthermore, an elevation in BT can enhance SANC excitability by regulating the conductance and gating kinetics of ionic channels [12,13], as well as the dynamics of intracellular Ca cycling [11]. Consequently, as BT rises, HR accelerates with a temperature coefficient ($Q_{10}$) of ~2 in rodents [14,15]. In adult mice, circadian rhythms of BT (CRBT), ranging from 36°C to 38°C, is characterized by a decline in BT during the day, followed by an increase in BT after waking up at night [16–18].

In addition to the ANS and CRBT, circadian rhythms in HR are also influenced by local circadian rhythmicity (LCR) (Fig 1; green dot) [19]. Prior experimental studies showed that circadian HR rhythms are lost in SCN-lesioned mice, while they are preserved in mice with local circadian disruption, suggesting a primary role of the ANS and a secondary role of LCR in regulating HR circadian rhythms [20–22]. However, under ANS blockade conditions, further studies demonstrate the significant contribution of LCR in promoting diurnal variations in the intrinsic HR (HR with no autonomic control) [22,23]. Notably, circadian rhythms in intrinsic HR can be eliminated with the blockade of hyperpolarization-activated cyclic nucleotide–

**Table 1. Glossary of circadian rhythms in HR.**

| Term | Explanation |
|---|---|
| Circadian rhythms | *Innate, roughly 24-hour biological cycles that regulate various physiological and behavioral processes in living organisms* |
| Robustness | *The heart's capability to sustain rhythmic HR consistently throughout the diurnal cycle, irrespective of external factors or conditions* |
| Performance | *The heart's capacity to achieve the maximum HR under conditions of increased demand or stress* |
| Flexibility | *The heart's capacity to produce a diverse range of HR throughout the day* |

gated channel ($I_{HCN}$) [19], confirming the essential role of $I_{HCN}$ in mediating LCR within the sinoatrial node (Fig 1; green dot).

Regulated jointly by the ANS, CRBT, and LCR (Fig 1), circadian rhythms in HR display intriguing properties of robustness, performance, and flexibility (Table 1). Despite changes in external conditions, they remain robust, ensuring the generation of rhythmic HR patterns throughout the day. They are capable of generating maximum HR (performance) in response to heightened demand or stress, often occurring after waking up. Moreover, they demonstrate flexibility, accommodating a broad range of pacing frequencies throughout a day, so that time-of-day cardiac outputs can be optimized under diverse physiological conditions, such as sleep/awake or inactive/active states. However, how ANS, CRBT, and LCR interact to facilitate circadian flexibility and performance in cardiac pacemaking function with robustness remains unclear. Furthermore, their vulnerability to adaptation under severe perturbations, such as aging (Fig 1; red dot), remains enigmatic [24–26].

To unravel these questions, the application of mathematical modeling and simulation proves invaluable in providing mechanistic insights into the non-linear behaviors of complex biological systems, e.g., mammalian circadian dynamics [27–31] and cardiac excitation patterns [32,33]. Earlier *in silico* studies have quantified the impact of circadian expression of potassium channel interacting protein-2 on shaping ventricular AP morphologies [34,35], and investigated the role of circadian rhythmicity of $I_{CaL}$ expression and function in the occurrence of early after-depolarizations in guinea pig ventricular cardiomyocytes and tissues [36]. However, previous computational studies of SANC have mainly emphasized the ionic interactions between MO and CO in the generation of spontaneous pacemaking activities [33,37]. There remains a paucity of *in silico* studies regarding the central and local circadian aspects of SANC automaticity.

In this study, we present a novel mathematical model that captures the intricate regulation of SANC by ANS, CRBT, and LCR in mice (Fig 1) (Table 2). The model accurately reproduces diverse circadian patterns as shown in previous experimental studies in adult and aged mice (19, 25, 38). Utilizing the model, we elucidate the specific roles of ANS, CRBT, and LCR in attaining circadian flexibility, performance, and robustness in SANC. Additionally, we quantitatively dissect SANC dysfunction during the aging process. Our findings reveal that ANS plays a pivotal role in promoting SANC flexibility and performance, with CRBT and LCR acting as primary and secondary boosters, respectively, to further enhance SANC flexibility and performance. However, during the aging process, while SANC flexibility and performance experience significant reductions, robustness remains mostly preserved. Specifically, the aging-related decline in PNA acts to restore SANC robustness while compromising flexibility, suggesting a potential trade-off strategy in the aging process. Our model simulations highlight a critical dimension for time-of-day interactions between ANS, CRBT, and LCR in cardiac pacemaking function and dysfunction.

**Table 2. Definitions of non-standard abbreviations.**

| Abbreviations | Definitions |
|---|---|
| ANS | *Autonomic nervous system* |
| AP | *Action potential* |
| BPM | *Beats per minute* |
| BT | *Body temperature* |
| cAMP | *Cyclic adenosine monophosphate* |
| CCh | *Carbachol* |
| CO | *$Ca^{2+}$ oscillator* |
| CRBT | *Circadian rhythm of BT* |
| FR | *Firing rates of a single sinoatrial nodal cell* |
| HR | *Heart rates* |
| $G_x$ | *Maximal conductance of ion channel x* |
| ISO | *Isoproterenol* |
| LCR | *Local circadian rhythmicity* |
| MO | *Membrane oscillator* |
| PNA | *Parasympathetic nervous activities* |
| $P_{up}$ | *Maximal rate of SERCA $Ca^{2+}$ pump* |
| PKA | *Protein kinase A* |
| SANC | *Sinoatrial nodal cells* |
| SCN | *Suprachiasmatic nucleus* |
| SERCA | *SR $Ca^{2+}$-ATPase* |
| SNA | *Sympathetic nervous activities* |
| SR | *Sarcoplasmic reticulum* |
| ZT | *Zeitgeber time* |

## Results

### Quantitative reconstruction of diverse circadian patterns of SANC FR in adult and aged mice

It has been suggested that in anesthetized mice without CRBT ($BT$ = 37˚C), PNA is more dominant in determining circadian HR variations compared to SNA (Fig 2A; blue vs yellow dashed lines) [38]. Specifically, the amplitude of circadian HR variations showed a significant reduction of 75% with a PNA blockade, whereas an SNA blockade resulted in a smaller reduction of 16%. Moreover, even after a complete ANS blockade, circadian HR variations persisted (Fig 2A; green dashed line) with an amplitude of ~2% [22,23]. Although this amplitude is minimal, this suggests that in addition to the ANS and CRBT, there can be other factors or mechanisms involved in regulating the circadian rhythmicity of HR in anesthetized mice, e.g., the daily changes in the intrinsic properties of the sinoatrial node arising from changes in the expression of $I_{HCN}$ (Fig 1; green dot) [19,38].

Our model simulations accurately recapitulated these variations in circadian rhythms of FR under conditions of control, SNA blockade, PNA blockade, and ANS blockade, when CRBT is not present (Fig 2B). Specifically, with ANS blockade, LCR (see Methods for more details) introduces a minor circadian amplitude of FR (Fig 2B; green line and Fig 2C and 2D), as experimentally observed in Fig 2A (green dashed line). While this amplitude marginally increases with PNA blockade (SNA+LCR) (Fig 2B; yellow line and Fig 2C and 2D), it considerably increases with SNA blockade (PNA+LCR) (Fig 2B; blue line and Fig 2C and 2D), consistent with experimental findings in (Fig 2A; yellow and blue dashed lines). Importantly, when

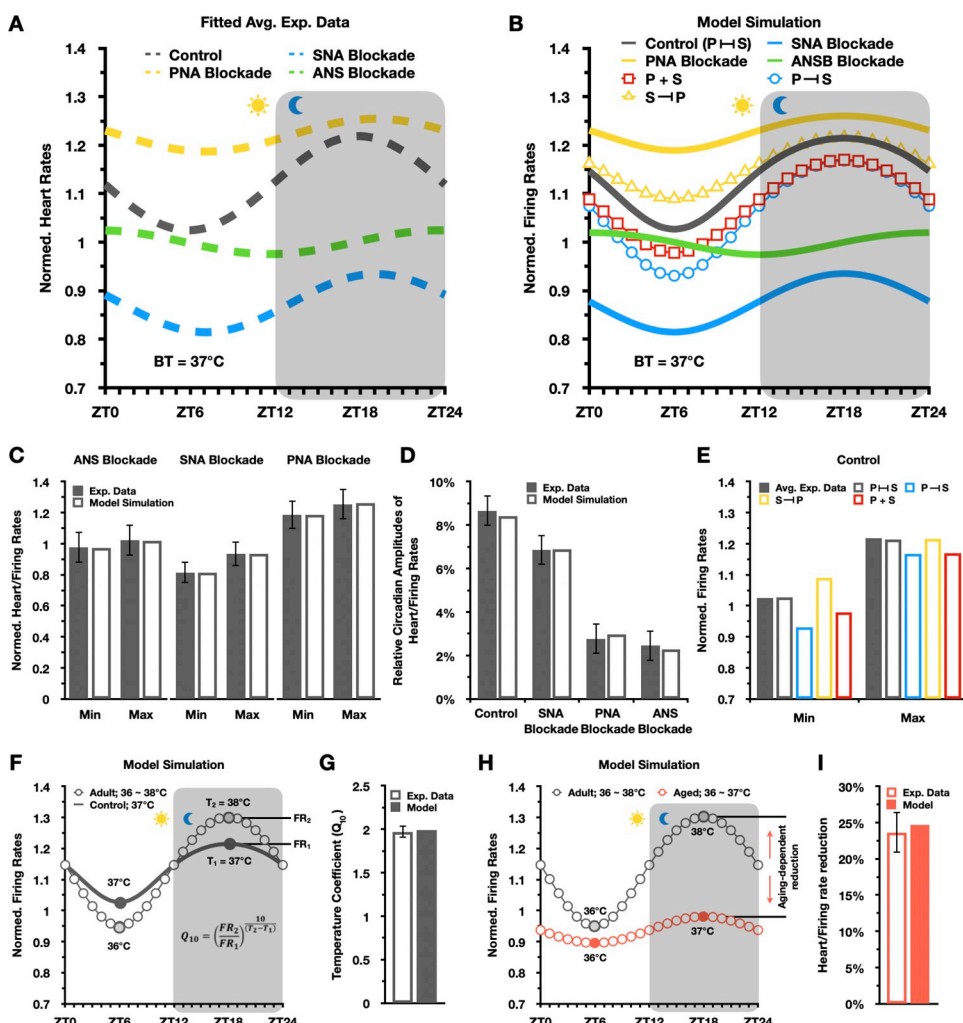

**Fig 2. Quantitative reconstruction of diverse circadian patterns in SANC FR under various conditions.** (A) Averaged circadian HR fluctuations in anesthetized mice ($BT = 37°C$) with 12-h light cycles before (control; grey dashed line) and after SNA blockade (blue dashed line), PNA blockade (yellow dashed line), and ANS blockade (green dashed line) [38]. Circadian fluctuations were fitted using a sine function as described in [38] and normalized to HR at ZT6 with ANS blockade. (B) Simulated circadian FR patterns in a single mouse SANC ($BT = 37°C$; normalized to FR at ZT6 with ANS blockade) closely recapitulate experimental findings [38]. (C-D) Model simulations accurately reproduce minimal and maximal time-of-day FR (C), and normalized circadian amplitudes of FR (D) under control, SNA blockade, PNA blockade, and ANS blockade conditions. The normalized amplitude values are obtained by dividing circadian FR amplitudes in beats per minute (BPM) by the mean time-of-day FR. (E) Either additive or unidirectional modulation effects alone are insufficient to accurately reproduce time-of-day FRs under the control conditions. (F-G) Circadian rhythmicity of $BT$ (36~38°C) enhances the circadian amplitude of FR under control conditions (grey dots) (F) with simulated $Q_{10} = 2$ in agreement with experimental studies (G) (15). (H-I) In aged model, simulated maximal FR reduction (25%; red dots) (H) is consistent with experimental findings (I) [25].

we implemented bidirectional modulation effects between PNA and SNA in the model, the circadian pattern in FR under the control condition was accurately simulated (Fig 2B; grey line and Fig 2C–2E) (see Methods for more details). Specifically, the PNA to SNA (P-S) modulation alone tends to increase the circadian amplitude of FR (Fig 2B; blue circle), while the SNA to PNA (S-P) modulation alone increases its baseline (Fig 2B; yellow triangle). As a result of the combined P-S and S-P effects, both the minimum and maximum time-of-day FR values are in close agreement with the experimental data (Fig 2E; grey filled vs. grey box). However,

in the absence of the PNA-SNA interactions (i.e., additive PNA and SNA), the baseline of the simulated circadian rhythm of normalized FR (Fig 2B; red box) was lower by ~5% in comparison to the experimental data (Fig 2A; grey dashed line). This suggests that the non-additivity of PNA and SNA is required to properly reconstruct the circadian patterns of SANC function. This finding is in agreement with the interactions between the sympathetic and parasympathetic branches of the ANS as accentuated antagonism [42–44].

These non-additive interactions could be crucial for gaining insights into the distinct mechanisms that underlie PNA in promoting circadian regulation of SANC FR, both in the absence and presence of SNA (Fig 2B; blue vs. grey lines). Specifically, the contribution of PNA on circadian amplitude of SANC FR via the direct activation of $I_{KACh}$ is paradoxically increased from 7% (Fig 2B; blue line) to 9% (Fig 2B; grey line) in the presence of SNA despite S-P inhibition. This might be primarily due to PNA's indirect time-of-day "breaking" effects on SNA, even with a weakened direct effect on $I_{KACh}$.

Next, expanding on our model with ANS and LCR (Fig 2B; grey line), we further incorporated CRBT (Fig 1) as a circadian function to reproduce the circadian patterns of $BT$ in adult mice, with an averaged time-of-day $BT$ of 37°C, and a circadian amplitude of 1°C. Then, we incorporated temperature-dependent factors to adjust gating kinetics and conductance of ion channels as previously described [10,45]. For ionic pumps, exchangers and intracellular ionic diffusion parameters, their temperature-dependent behaviors were simulated by scaling their maximal capacity or original value with $Q_{10}$. Furthermore, we modeled the temperature-dependent activities of SNA and PNA by introducing linear adjustments based on experimental measurements [14] (see Methods for details). With CRBT, simulated circadian amplitude of SANC FR was further enhanced (Fig 2F; grey dots), yielding a $Q_{10}$ value of 2 as previously reported in experimental studies (Fig 2G) [14,15]. We refer this model with CRBT, ANS, and LCR as adult model throughout this work.

Furthermore, based on our adult model, we developed and validated an aged model that incorporates aging-dependent alterations [25,39–41]. Specifically, aging-dependent ion channel remodeling in $I_{CaT}$, $I_{CaL}$, $I_{HCN}$, $I_{NaK}$, $I_{NCX}$, $I_{Kr}$, $I_{Ks}$ were modeled by scaling their maximal conductance and gating kinetics based on experimental findings [25,39,40,46]. Aging-associated reduction in the expression level of $Ca^{2+}$ cycling proteins, e.g., SERCA ($J_{up}$) and Ryanodine receptor ($J_{Rel}$), was modeled by adjusting their maximal activities and kinetics in alignment with experimental measurements [40]. Age-related PNA impairment was simulated by a major reduction in PNA based on earlier experimental studies [41]. Additionally, CRBT disruption was modeled by reducing the baseline and amplitude of CRBT, consistent with experimental findings in aged mice [17,26,41,47] (see Methods for details). As a result, the simulated circadian amplitude of SANC FR was largely dampened in aged model (Fig 2H; red dots), with a major reduction (25%) of maximal time-of-day FR as experimentally measured (Fig 2I) [25].

## The alliance of ANS, CRBT, and LCR is essential in optimizing the time-of-day SANC function in adult mice

Our model successfully recapitulated diverse circadian patterns in SANC pacemaking function (Fig 2). Subsequently, we utilized the model to investigate how CRBT, ANS and LCR interact to regulate robustness, performance, and flexibility in SANC pacemaking function. To achieve this, we simulated CO-MO parameter space maps at ZT6 and ZT18 under control conditions in adult model (Fig 3A). Specifically, each CO-MO parameter space map was generated by reducing the maximal rate of SERCA ($P_{up}$) (a CO parameter; along the vertical axis) and the maximal conductance of $I_{CaL}$ and $I_{CaT}$ (MO parameters; along the horizontal axis) from their

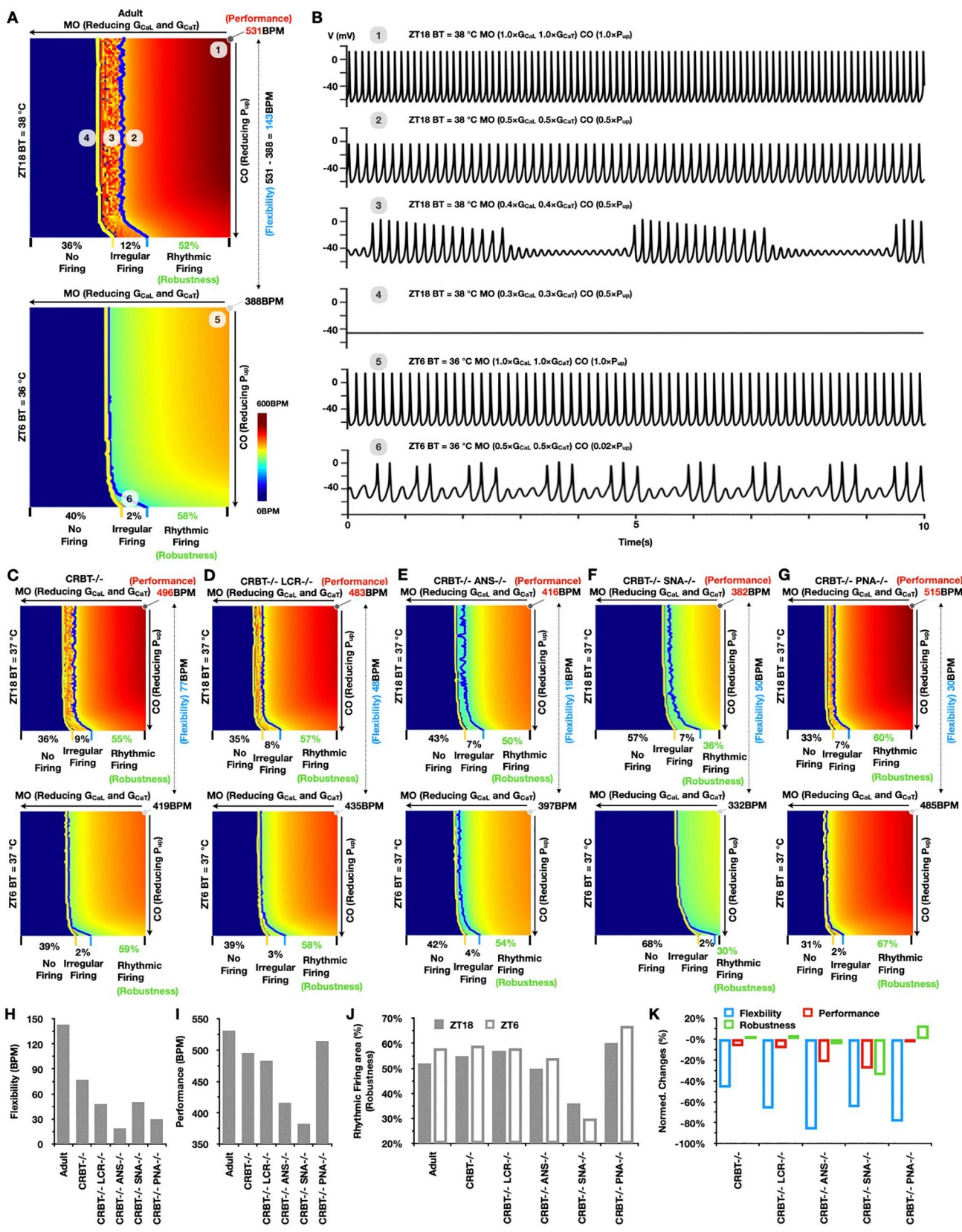

**Fig 3. The alliance of ANS, CRBT and LCR is essential to achieve circadian flexibility and performance while preserving robustness in SANC automaticity.** (A) CO-MO parameter space maps color-coded by FR in BPM at ZT6 and ZT18 under control conditions in adult model. (B) Steady-state SANC membrane potential oscillations using parameter settings sampled from panel (A; labeled 1 to 6). (C-G) CO-MO parameter space maps at ZT6 and ZT18 under (CRBT-/-) (C), (CRBT-/-; LCR-/-) (D), (CRBT-/-; ANS-/-) (E), (CRBT-/-; SNA-/-) (F), and (CRBT-/-; PNA-/-) (G) conditions. (H-J) Quantification of SANC flexibility (H), performance (I), and robustness (J) under various conditions. (K) Normalized changes in SANC flexibility, performance, and robustness compared to the control conditions in adult model.

original values to zero using a step size of 1%, resulting in a total of $100 \times 100$ simulations. The original values for these CO and MO parameters were defined with $P_{up}$ = 0.04 mM/ms, $G_{CaL}$ = 0.018 nS/pF and $G_{CaT}$ = 0.013956 nS/pF as previously described [48]. The steady-state FR in BPM was computed for each simulation and color-coded to create the map.

In the CO-MO parameter space maps, borders between no firing, irregular firing, and rhythmic firing regions were denoted by yellow and blue lines, respectively (Fig 3A). Representative steady-state SANC membrane potential oscillation traces (Fig 3B) were sampled from Fig 3A (labeled 1 to 6) to illustrate distinct membrane excitation patterns under various parameter settings of CO and MO, ranging from fast (Fig 3A; 1) to slow ((Fig 3A; 5) rhythmic firing, from irregular (Fig 3A; 3,6) to no firing (Fig 3A; 4).

Because rhythmic firing regions represent the parameter space where SANC maintains rhythmic pacemaking, we quantified SANC robustness (Table 1) as the percentage area of rhythmic firing regions (Fig 3A). Under control conditions, averaged time-of-day SANC robustness is 55% (58% and 52% at ZT6 and ZT18, respectively). Such time-of-day differences in SANC robustness may be attributable to the diminished region of irregular firing (2%) at ZT6, compared to 12% at ZT18. SANC performance was quantified as the maximal time-of-day SANC FR, representing the peak FR reached during the circadian cycle, which is 531 BPM in the control. In addition, SANC flexibility was quantified as the difference between maximal (ZT18) and minimal (ZT6) time-of-day SANC FR with control CO and MO parameter values, reflecting the ability of SANC to adjust its FR over the circadian cycle. SANC flexibility under control conditions is 143 BPM, which is the difference between 531 BPM (ZT18) and 388 BPM (ZT6) (Fig 3A).

To quantitatively dissect the specific roles of CRBT, ANS and LCR in promoting SANC pacemaking function, we simulated CO-MO parameter space maps under various conditions: (CRBT-/-), (CRBT-/-; LCR-/-), (CRBT-/-; ANS-/-), (CRBT-/-; SNA-/-), and (CRBT-/-; PNA-/-) (Fig 3C–3G). Then we quantified the flexibility (Fig 3H), performance (Fig 3I) and robustness (Fig 3J) under these conditions and how much they are changed (Fig 3K). Without CRBT, both SANC flexibility (77BPM) and performance (496BPM) are significantly decreased; yet, averaged time-of-day robustness is slightly enhanced (57%). When LCR is additionally blocked, SANC flexibility (48BPM) and performance (483BPM) are further reduced, with averaged SANC robustness of 57.5%. Similarly, without CRBT and ANS, SANC flexibility (19BPM) and performance (416BPM) are substantially impaired, with a minor reduction in averaged SANC robustness (52%). Additionally, without CRBT and SNA, SANC performance (382BPM) and robustness (averaged at 33%) are preferentially impaired over flexibility (50BPM). On the other hand, without CRBT and PNA, both performance (515BPM) and robustness (averaged at 63.5%) are markedly enhanced, while SANC flexibility (30BPM) is further reduced. These findings suggest that SNA preferentially enhances performance and robustness, while PNA inclines to amplify flexibility at the expense of both performance and robustness (Fig 3K). As a result, ANS promotes both SANC flexibility and performance, without compromising robustness. In addition, CRBT and LCR may act as primary and secondary boosters for SANC flexibility and performance (Fig 3K).

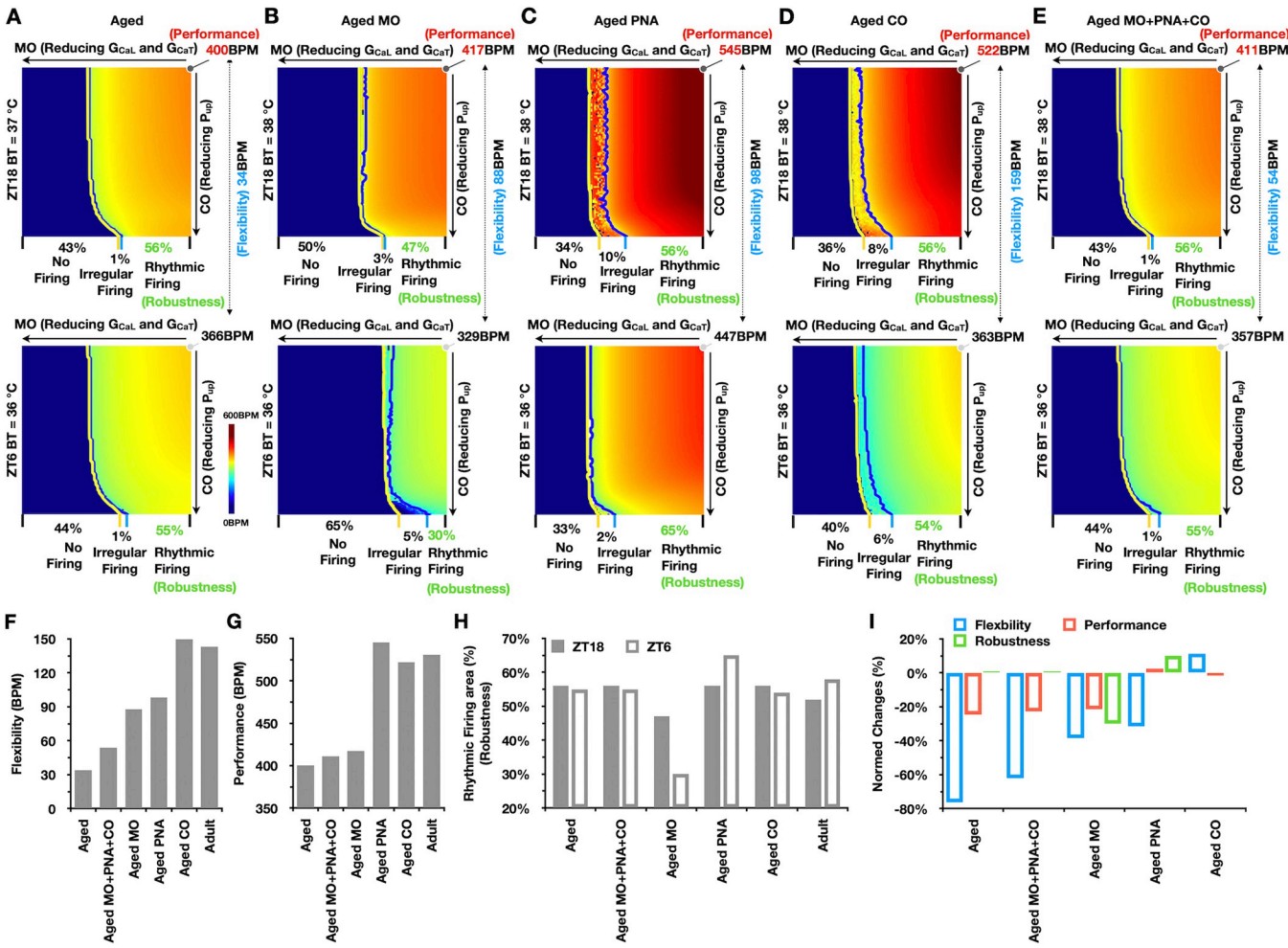

**Fig 4. Quantitative dissection of SANC pacemaking dysfunction in aging.** (A-E) CO-MO parameter space maps at ZT6 and ZT18 under aged (A), aged MO (B), aged PNA (C), aged CO (D), and aged MO+PNA+CO (E) conditions. (F-H) Quantitative differences in flexibility (F), performance (G), and robustness (H) under various aging conditions. (I) Normalized changes in SANC flexibility, performance, and robustness compared to the control conditions in adult model.

### Quantitative dissection of SANC pacemaking dysfunction in aged mice

Using aged model (Fig 2H; red dots), we simulated CO-MO parameter space maps at ZT6 and ZT18 under control conditions (aged; Fig 4A). According to the map, both SANC flexibility (34BPM) and performance (400BPM) were significantly impaired; yet, surprisingly, SANC robustness was well preserved (averaged at 55.5%) (Fig 4F–4H) compared to the adult model (Fig 3A).

To understand how each aging-dependent alteration collectively contributes to the deterioration in SANC pacemaking function during aging, we conducted additional simulations of CO-MO parameter space maps at ZT6 and ZT18 by incorporating aging-dependent MO remodeling (aged MO; Fig 4B), PNA impairment (aged PNA; Fig 4C) and CO alteration (aged CO; Fig 4D) individually, and their combination (aged MO+PNA+CO; Fig 4E). With aged MO (Fig 4B), SANC flexibility (88BPM), performance (417BPM), and robustness (averaged at 38.5%) were all substantially impaired compared to the adult model (Fig 4F–4H). On the other hand, with aged PNA (Fig 4C), SANC performance (545BPM) and robustness (averaged at

60.5%) were largely enhanced instead, with a reduction in SANC flexibility (98BPM) (Fig 4F–4H). Furthermore, with aged CO (Fig 4D), SANC performance (522BPM) and flexibility (159BPM) were moderately reduced and enhanced, respectively, while robustness (averaged at 55%) was mostly unchanged (Fig 4F–4H). With combined aging effects associated with MO, PNA, and CO (aged MO+PNA+CO; Fig 4E), SANC flexibility (54BPM) and performance (411BPM) were further reduced; yet, there was a full restoration of SANC robustness (averaged at 55.5%) compared to the adult model (Fig 4F–4H). When aged CRBT is added to the aged MO+PNA+CO (aged; Fig 4A), SANC flexibility (34BPM) and performance (400BPM) are further reduced, while robustness is unchanged (averaged at 55.5%) (Fig 4F–4H). These model simulations suggest that while aging-dependent MO remodeling in aging impairs all aspects of SANC pacemaking function, aging-dependent changes in PNA and CO may counteract MO remodeling in aging to buffer its damaging effects in SANC (Fig 4I). In addition, CRBT disruption further promotes aging-dependent reduction in both flexibility and performance of SANC (Fig 4I).

## Distinct mechanisms underlying circadian patterns in SANC FR in adult and aged mice

To dissect key mechanisms underlying circadian patterns in SANC, we assessed the relative contributions of each module in the model to circadian flexibility and performance of SANC. Specifically, we quantified how much the flexibility or performance of SANC was perturbed after fully inhibiting ANS, CRBT, or LCR in the adult model (Fig 5A) and aged model (Fig 5B). Furthermore, as the role of intracellular $Na^+$ ($[Na]_i^+$) accumulation in regulating cardiac excitation patterns has been previously reported [49–51], we also clamped the time-of-day $[Na]_i^+$ content at its steady-state concentration at ZT6 to evaluate its potential role in regulating the flexibility and performance of SANC.

In adult model (Fig 5A), while ANS inhibition significantly reduces the circadian flexibility (-78%) and performance (-20.2%) of SANC, the effects of CRBT or LCR inhibition are secondary in reducing circadian flexibility (-46.5% and -6.7%, respectively) and performance (-19.2% and -2.8%, respectively). Furthermore, $[Na]_i^+$ clamping exerts an enhancing effect on both SANC flexibility (-10.6%) and performance (-2.9%), attributed to a higher $[Na]_i^+$ content at faster FR. However, in aged model, different mechanisms underlying the circadian flexibility and performance of SANC were observed (Fig 5B). While the effects of ANS inhibition remain

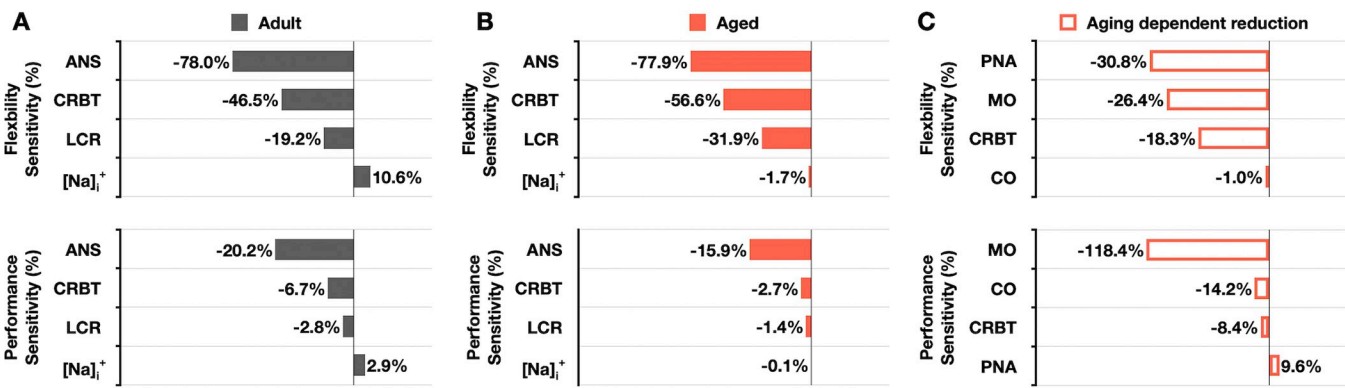

**Fig 5. Distinct mechanisms underlying circadian patterns of SANC FR in adult and aged mice.** (A-B) Relative contributions to SANC flexibility and performance are quantified by selective inhibition of each module in the adult (A) and aged (B) models. (C) Relative contributions to aging-dependent reduction in SANC flexibility and performance are quantified by selective inhibition of each aging-dependent change in the aged model (red box).

most sensitive, leading to a reduction in SANC flexibility (-77.9%) and performance (-15.9%), the effects of CRBT and LCR inhibition are enhanced, causing a reduction in flexibility (-56.6% and -31.9%, respectively) compared to the adult model (Fig 5A and 5B). This enhanced effect may be attributable to aging-dependent PNA impairment. In addition, the effect of $[Na]_i^+$ clamping becomes diminished and reversed in the aged model (Fig 5B) compared to the adult model (Fig 5A).

To further quantify mechanisms underlying aging-dependent reduction in SANC flexibility and performance in aged model, we additionally quantified the extent to which either flexibility or performance of SANC was perturbed by fully inhibiting MO remodeling, PNA impairment, CO alteration and CRBT disruption using the aged model (Fig 5C). The reduction of SANC flexibility in aging is mostly dampened by the inhibition of PNA impairment (-30.8%) and MO remodeling (-26.4%), with a secondary contribution of the inhibition of CRBT disruption (-18.3%), and a negligible effect by the inhibition of CO alteration (-1%). However, the reduction of SANC performance in aging is predominantly weakened by the inhibition of MO remodeling (-118.4%), with secondary contributions of the inhibition of CO alteration (-14.2%) and CRBT disruption (-8.4%). On the other hand, the reduction of SANC performance in aging is promoted by the inhibition of PNA impairment (9.6%).

## Discussion

It is known that circadian rhythms in HR can arise from either the master circadian clock located in the SCN or the local circadian clock in the heart [2,4]. Previous experimental studies have suggested that the 24h HR rhythm is primarily governed by the SCN through the ANS [4,19,52,53]. However, instead of solely focusing on determining which one, the master or local circadian clock, dominates, our study was motivated to understand the necessity of having two circadian clocks to regulate cardiac pacemaking function and to explore their specific roles. Earlier computational studies have primarily focused on the dynamic interactions between $Ca^{2+}$ and membrane oscillations in generating spontaneous pacemaking activities in SANC [37,54]. However, the circadian aspects of SANC pacemaking function over the course of the day have not been established [55].

Our study sheds light on a critical yet understudied dimension of time-of-day interactions between master and local clocks in cardiac pacemaking function and dysfunction, by developing a model of ANS, CRBT, and LCR in SANC to recapitulate diverse circadian patterns in both adult and aged mice. Leveraging this model, we elucidated the distinctive roles of ANS, CRBT, and LCR as a dominant amplifier, a primary booster, and a secondary booster, respectively, in achieving circadian flexibility and performance with robustness in SANC. As illustrated in Fig 6A, departing uphill from the baseline state (grey circle), the introduction of LCR (green circle) moderately enhances SANC flexibility and slightly increases performance without affecting robustness. SNA addition (yellow circle) primarily enhances SANC robustness and performance, while PNA (blue circle) promotes SANC flexibility at the expense of both performance and robustness. The combination of ANS and LCR (grey circle) significantly enhances both flexibility and performance without compromising robustness. The final incorporation of CRBT (grey dot; adult) further augments both flexibility and performance.

However, under aging conditions, the cooperative alliance of ANS, CRBT, and LCR is disrupted, resulting in a substantial reduction in the performance and flexibility of SANC. Surprisingly, SANC robustness remains well-preserved. As shown in Fig 6B, departing downhill from the adult state (grey dot), aging-dependent ion channel remodeling (blue circle) undermines all aspects of the SANC function. However, aging-dependent PNA impairment (yellow circle) enhances both SANC performance and robustness with a trade-off in flexibility,

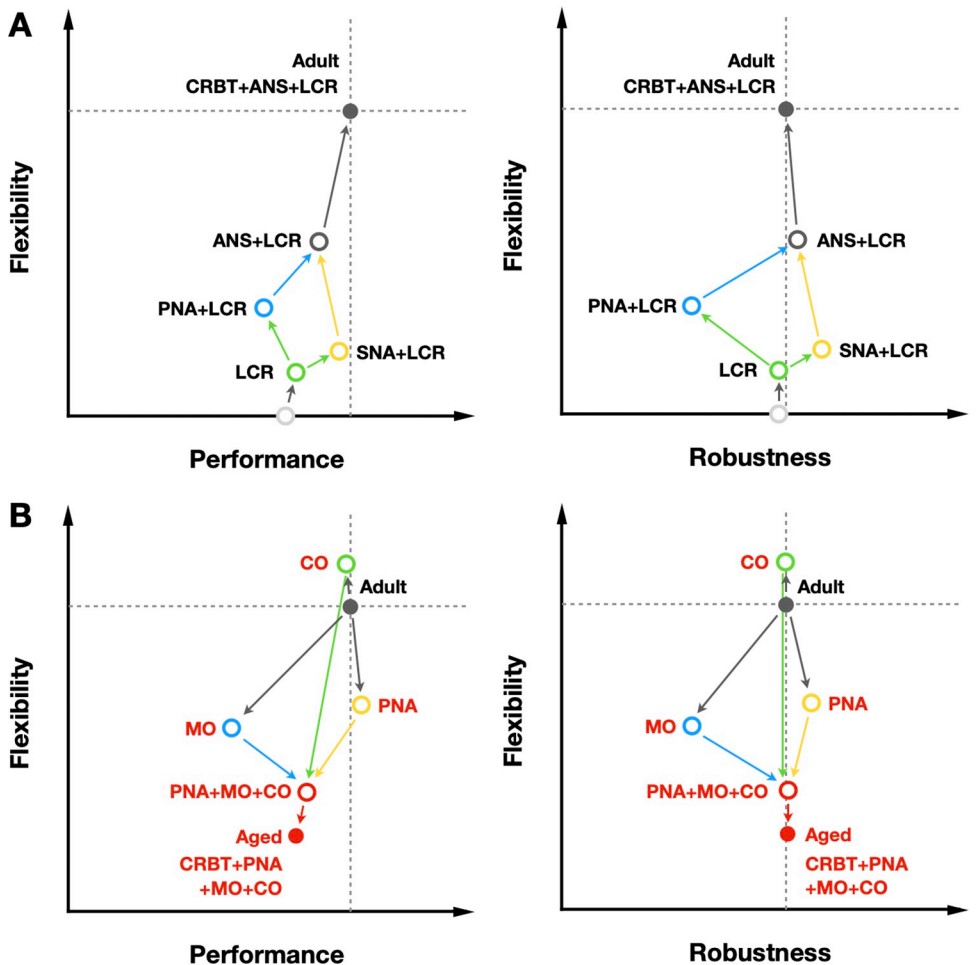

**Fig 6. Illustrative trajectories in the parameter space of SANC flexibility, performance and robustness.** (A) Uphill trajectories show the transition from baseline (grey circle) to adult (CRBT + ANS + LCR; grey dot) model states. (B) Downhill trajectories demonstrate the transition from adult (grey dot) to aged model states (red dot).

countering the effects of MO remodeling. Moreover, aging-dependent Ca²⁺ handling alterations (green circle) introduce a minor reduction and enhancement in performance and flexibility respectively, without affecting robustness. The combined effects of aging-dependent changes in MO, PNA, and CO (red circle) further reduce SANC flexibility, yet fully restore robustness compared to the adult state (grey dot). With the final addition of CRBT disruption in aging, SANC flexibility and performance are both further reduced without affecting robustness. Comparing Fig 6A and 6B, it is intriguing to observe the well-preserved robustness evident in both adult and aged states [56,57]. In adult model, SNA counters PNA, ensuring the preservation of robustness. In aged model, the impairment of PNA works against ion channel remodeling to restore SANC robustness and maintain basal cardiac function.

These mechanistic insights governing the circadian regulation of SANC pacemaking could be a critical step towards a comprehensive understanding of the circadian control of cardiac function and dysfunction [2,4]. For instance, the circadian variation in SANC automaticity alone can be an important factor in determining arrhythmogenic vulnerabilities in ventricular excitation, due to rate-dependent properties of ventricular myocytes [58]. While an elevated HR may promote the occurrence of electric and Ca²⁺ alternans in ventricular tissue [59], a

slow HR could contribute to the development of ventricular repolarization abnormalities [49,60], potentially leading to life-threatening ventricular arrhythmias [61]. In addition, the role of LCR in regulating cardiac function could be tissue-specific. While LCR may act as a secondary booster for SANC function, earlier experimental studies suggest that in ventricular myocytes, LCR can serve as a buffer against QT interval prolongation and circadian changes in neurohormonal signaling [4]. Mathematical models that quantitatively describe tissue-specific circadian regulation of the heart, may serve as invaluable tools in dissecting the chronobiological mechanisms that underlie diverse circadian patterns of cardiac arrhythmic phenotypes [2].

The timing of drug administration (e.g., morning or afternoon) can significantly impact its effectiveness and side effects due to circadian rhythms [62,63]. For instance, taken in the morning, melatonin can delay sleep onset, while evening administration can promote it [64]. Similarly, simvastatin, a cholesterol-lowering medication, exhibits greater efficacy when taken at night [63]. Even the efficacy and toxicity of cancer chemotherapy strongly depend on dosing time [65,66]. Chronotherapy of cardiac and vascular disease has been recently reported [67]. Because mathematical models have played critical roles to investigate the chronotherapy [68–71], it would be interesting future work to investigate cardiac chronotherapy using the mathematical model developed in this study. For example, melatonin exhibits antiarrhythmic effects linked to its role in promoting intercellular coupling and reducing heterogeneity of ventricular repolarization. However, chronic melatonin supplementation with personalized chronotherapy remains to be established [72]. In addition, while hERG channel inhibition and the Comprehensive In Vitro Proarrhythmia Assay are widely used to assess drug-induced arrhythmic events, the circadian aspects of drug-induced arrhythmic risks have not been considered [73]. Our model may help to optimize chronotherapeutic strategies and accelerate the identification of novel therapeutic targets within circadian clocks for the prevention and treatment of cardiac arrhythmias.

We have made several assumptions for simplicity in developing our model that may impact the generalizability and interpretation of our model simulations. Our adult model was calibrated to experimental data in anesthetized mice [38] assuming limited circadian activities in SNA [26], which may potentially underestimate the role of SNA in SANC flexibility in free-moving mice living in a thermoneutral environment [74]. Our aged model was parameterized based on a cross-sectional experimental dataset in >32 months old mice, thus lacking the capacity to capture functional trajectories throughout the aging process. We modeled LCR as a diurnal variation in the expression level of $I_{HCN}$, but it might not be the only mechanism underlying LCR in SANC. For example, a day/night difference has been reported in several $K^+$ channels and in $Ca^{2+}$/calmodulin-dependent protein kinase II delta expression [38]. We modeled CRBT as a simplified sinus function that peaks at ZT18, while actual experimental recordings of CRBT could feature additional complexities with surges in $BT$ immediately after waking up and before sleep, and potential phase advances in aged mice [17]. Inherited from the baseline model [48,51,75], specific molecular identities of several ion channels (e.g., $I_{st}$, $I_{Nab}$, $I_{Cab}$) remain unknown, and a biochemical description of beta-adrenergic and cholinergic signaling pathways and their interactions is absent. In addition, our model is limited by the lack of a detailed transcription-translation feedback loop model for LCR in SANC, owing to the scope of this study and the limited availability of experimental data. Additional model development or experimental studies would be helpful to further advance our understanding of circadian SANC function and dysfunction, e.g., non-additive interactions between SNA and PNA[6], the mechanistic coupling between the ANS and LCR [76], trajectory of functional aging [77], and entrainment dynamics between the master circadian clock and LCR during jet lag or with shift work disorder [78].

## Methods

### Model development

Autonomic, electrophysiologic, and circadian properties of cardiac function are species-dependent [26,79]. Given the availability of experimental data in mice, our model is constructed to be mouse-specific.

We used the mathematical model of mouse SANC AP developed by Ding et al. [48,51], based on the original work of Kharche et al. [75], as our baseline model. All baseline model definitions, equations, and parameter settings were unchanged from those previously described and implemented by Ding et al. [48]. Due to the absence of circadian regulation in the baseline model, we extended it to quantitatively describe circadian variations in ANS, CRBT, and LCR for both adult and aged mice. Detailed descriptions of model equations are provided in the following section, along with all parameter values and settings listed in S1 Table.

### Incorporation of LCR into the baseline model

First, without considering CRBT, LCR was introduced and calibrated into the baseline model as a circadian variation in the $I_{HCN}$ (Fig 1; green dot) to generate a weak circadian amplitude (~2%) in SANC FR as observed in adult mice under full ANS blockade with $BT$ = 37˚C (Fig 2A; green dashed line) [19,38]. Specifically, to introduce circadian variations in $I_{HCN}$ into the baseline model with ANS blockade, $I_{HCN}$ was modified from its original form $I_{HCN} = I_{HCN,Na} + I_{HCN,K}$ as follows:

$$I_{HCN} = \left(1 + A_{LCR} \times \cos\left(\frac{2\pi T}{T_{LCR}}\right)\right) \times \left(I_{HCN,Na} + I_{HCN,K}\right),$$

where $A_{LCR}$ is the amplitude of LCR, $T_{LCR}$ is the period of LCR, $T$ is the current ZT in hours, $I_{HCN,Na}$ and $I_{HCN,K}$ are the Na$^+$ and K$^+$ specific HCN channel currents, respectively [48]. It should be noted that, while experimental recordings of HR with ANS blockade [38] show troughs at ZT12, experimental measurements of $I_{HCN}$ currents [19] show troughs at ZT6. We speculated that this might be attributable to a phase shift in LCR due to a lack of external cues from the ANS. For this, when the ANS is present, $I_{HCN}$ was modeled as follows:

$$I_{HCN} = \left(1 + A_{LCR} \times \cos\left(\frac{2\pi \times (T + 6)}{T_{LCR}}\right)\right) \times \left(I_{HCN,Na} + I_{HCN,K}\right).$$

### Incorporation of the circadian regulation of ANS into the baseline model

Then, circadian PNA (Fig 1; blue dot) was added and calibrated as a circadian function of carbachol (*CCh*) concentrations, to reproduce the circadian amplitude (~7%) effects based on experimental data under SNA blockade conditions with $BT$ = 37˚C (Fig 2A; blue dashed line) [38]. For this, *CCh* was modified from its original form $CCh = CCh_{basal}$ [48] as follows:

$$CCh = CCh_{basal} + A_{CCh} \times \cos\left(\frac{2\pi \times (T - 6)}{T_{CCh}}\right),$$

where $CCh_{basal}$ is the basal concentration of *CCh*, $A_{CCh}$ is the circadian amplitude of *CCh*, $T_{CCh}$ is the period of *CCh* variations, and $T$ is the current time in ZT hours. This circadian variation in *CCh* leads to the circadian variation in the muscarinic K$^+$ current ($I_{KACh}$) because $I_{KACh}$ is regulated by *CCh* in the original baseline model as follows (Fig 1; blue dot) [48]:

$$I_{KACh} = G_{KAch} \times w(V, CCh, t) \times (V - E_K),$$

where $G_{KACh}$ is the maximal conductance of $I_{KACh}$, $V$ is the membrane potential in mV, $t$ is current time in ms, $w(V,CCh,t)$ is the voltage- and $CCh$-dependent channel activation function, and $E_K$ is the K$^+$ equilibrium potential in mV.

Furthermore, SNA (Fig 1; yellow dot) was implemented to mimic isoproterenol ($ISO$) effects by modifying the kinetics and/or conductance of $I_{st}$, $I_{Na1.1}$, $I_{CaT}$, $I_{CaL}$, $I_{K1}$, $I_{Ks}$, $I_{Kr}$, $I_{to}$, and the activities of $J_{Rel}$ and $J_{up}$, using parameter settings as previously described (see S1 Table for more details) [48]. The administration of $ISO$ was scaled to align with experimental measurements of HR acceleration under PNA blockade conditions ($BT = 37°C$) (Fig 2A; yellow dots) [38]. A combination of LCR and $ISO$ administration is enough to generate a circadian amplitude (~3%) in SANC FR as measured in the experiments [38], allowing little room for additional in-phase circadian variations in SNA. Therefore, SNA in this study was modeled with no circadian variations when CRBT is not present ($BT = 37°C$). This modeling choice aligns with earlier experimental findings indicating a high sympathetic drive in mice, potentially limiting additional time-of-day variations to maintain a normal core temperature (37°C) under standard laboratory conditions (20°C) [26,74].

When SNA and PNA were both present, bidirectional modulation effects between PNA and SNA (P-S and S-P) (Fig 2B) were formulated and calibrated to account for the non-additivity effects [6,42,80,81] and to be consistent with experimental measurements (Fig 2A; grey dots) [38]. The non-additivity effects with $ISO$ and $CCh$ were implicitly modeled (e.g., via cAMP-PKA dependent pathways or crosstalk between sympathetic and parasympathetic nervous systems) by introducing the $ISO$ effects at a given concentration of $CCh$ as follows:

$$E_{ISO} = e_{iso} \times \frac{k_{p\_s}^h}{k_{p\_s}^h + CCh^h},$$

where $e_{iso}$ is the stand-alone $ISO$ effects as previously described [48], $k_{p\_s}$ is the P-S modulation constant, and $h$ is the Hill coefficient. Furthermore, $CCh$ effects with $ISO$ were introduced as

$$E_{CCh} = e_{cch} \times k_{s\_p},$$

where $e_{cch}$ is the $CCh$ effects alone as previously described [48], and $k_{s\_p}$ is the S-P modulation scaling factor (e.g., to account for the cross-talk between SNA and PNA [6,42,80,81]).

## Incorporation of CRBT into the baseline model

CRBT (Fig 1) was added and calibrated as a circadian function of $BT$ to reproduce the circadian patterns of $BT$ in adult mice (with averaged time-of-day $BT$ of 37°C and a circadian amplitude of 1°C) [38] (Fig 2F; grey dots). For this, $BT$ was modified from its original form $BT = BT_\alpha$ [48] as follows:

$$BT = BT_a + A_{BT} \times \cos\left(\frac{2\pi \times (T + 6)}{T_{BT}}\right),$$

where $BT_a$ is the reference $BT$, $A_{BT}$ is the circadian amplitude of $BT$, $T_{BT}$ is the period of $BT$ variations, and $T$ is the current time in ZT hours.

To account for the effects of CRBT for all ion channels (excluding ionic pumps and exchangers) in our model, we introduced two temperature-dependent factors $\varphi(BT)$ and $\eta(BT)$, to scale gating kinetics and conductance of ion channels respectively, as previously described in [10,45] as follows:

$$\varphi(BT) = Q_{10}^{\frac{BT - BT_a}{10}},$$

$$\eta(BT) = 1 + B(BT - BT_a),$$

where $BT$ is the current body temperature, $BT_a$ is the reference body temperature, $B$ is a scaling coefficient. For example, the temperature dependence of ion channel $x$ was modeled as follows:

$$\tau_{x,BT} = \frac{\tau_x}{\varphi(BT)}$$

$$G_{x,BT} = \eta(BT)G_x,$$

where $G_x$ and $\tau_x$ are the original conductance and gating time constant of ion channel $x$ as in the baseline model, $G_{x,BT}$ and $\tau_{x,BT}$ are the conductance and gating time constant of ion channel $x$ at a given $BT$.

For ionic pumps, exchangers, and intracellular ionic diffusion parameters, their temperature-dependent behaviors were modeled by scaling their maximal activity or original value (if a constant) with $\varphi(BT)$ For SNA and PNA, it has been reported in earlier experiments [14] that around normal $BT$ (e.g., 37˚C), as $BT$ increases, there is a linear increase and decrease in SNA and PNA, respectively, to promote HR acceleration. Thus, the temperature-dependent behaviors of SNA and PNA were modeled by scaling their activities with $\eta(BT)$ based on experimental findings [14].

## Modeling aging-dependent alternations

Aging-dependent ion channel remodeling (Fig 1; red dot) was calibrated to be consistent with electrophysiological measurements by Larson et al. [25], Tellez et al. [39] and Liu et al. [40] as follows:

$$I_{CaT,aging} = a_{cat} \times I_{CaT,}$$

$$I_{CaL,aging} = a_{cal} \times I_{CaL},$$

$$I_{HCN,aging} = a_{hcn} \times I_{HCN},$$

$$I_{NaK,aging} = a_{nak} \times I_{NaK},$$

$$I_{NCX,aging} = a_{ncx} \times I_{NCX},$$

$$I_{Kr,aging} = a_{hcn} \times I_{Kr},$$

$$I_{Ks,aging} = a_{hcn} \times I_{Ks}$$

where $a_{cat}$, $a_{cal}$, $a_{hcn}$, $a_{nak}$, $a_{ncx}$, $a_{kr}$, $a_{ks}$ are the scaling factors for $I_{CaT}$, $I_{CaL}$, $I_{HCN}$, $I_{NaK}$, $I_{NCX}$, $I_{Kr}$, and $I_{Ks}$ under aging conditions, respectively. Moreover, the aging-dependent shift of the activation midpoint of $I_{HCN}$ was modeled as the following:

$$V_{HCN,aging} = V_{HCN} + v_{hcn},$$

where $V_{HCN}$ is the activation midpoint of $I_{HCN}$ in adult model, and $v_{hcn}$ is the activation shift of $I_{HCN}$ in aged model.

Aging-dependent $Ca^{2+}$ cycling dysfunction (Fig 1; red dot) in SERCA ($J_{up}$) and Ryanodine receptor Ca release ($J_{Rel}$) were calibrated according to previous experimental measurements by Liu et al. [40] as follows:

$$J_{up,aging} = a_{up} \times J_{up}$$

$$J_{Rel,aging} = a_{rel} \times J_{Rel}$$

where $a_{up}$, $a_{rel}$ are the scaling factors for $J_{up}$, $J_{Rel}$ under aging conditions, respectively. Moreover, the aging-dependent increase in the ratio between phospholamban and SERCA [40] was modeled by an increase to the original value of cytoplasmic $Ca^{2+}$ sensitivity coefficient ($K_{mf}$) in the baseline model [48] as follows:

$$K_{mf,aging} = a_{kmf} \times K_{mf}$$

where $a_{kmf}$ is the scaling factor for $K_{mf}$ under aging conditions.

In addition, aging-related PNA impairment (Fig 1; red dot) was introduced by a major reduction in PNA to be consistent with previous experimental studies [41] as follows:

$$CCh_{aging} = a_{cch} \times CCh$$

where $a_{cch}$ is the scaling factor of $CCh$ under aging conditions. SNA was assumed to be unchanged in aged mice as reported by Larson et al. [25].

Moreover, aging-dependent CRBT disruption was modeled by a reduction in both averaged time-of-day $BT$ and circadian amplitude of $BT$ in aged mice as previously reported [17]. Specifically:

$$BT_{aging} = BT_{a,aging} + A_{BT,aging} \times \cos(\frac{2\pi \times (T + 6)}{T_{BT}}),$$

where $BT_{a,aging}$ is the reference $BT$ in aged model, and $A_{BT,aging}$ is the circadian amplitude of $BT$ in aged model.

## Simulation protocols

A 12h:12h light/dark lighting regime was implemented as in the experimental studies [19,38]. FR (Figs 2–5) was calculated as the averaged FR of the last 10 seconds of simulation after reaching steady-state, with model implementation and initial conditions previously described by Ding et al. [48]. Each CO-MO parameter space map (Figs 3 and 4) [37] was reproduced by reducing $P_{up}$ (CO) and the maximal conductance of $I_{CaL}$ and $I_{CaT}$ (MO) from their original values [48] to zero, using a step size of 1%, resulting in a total of $100 \times 100$ simulations. Regarding the choice of parameters to represent CO and MO, respectively, we chose $P_{up}$ to represent CO as often used in previous modeling studies [37,54,82]. Both $I_{CaL}$ and $I_{NCX}$ have been used to represent MO in earlier studies [37,54,82]. Given the importance of $I_{CaT}$ in mouse SANC [51,83], we chose to use $I_{CaL}$ and $I_{CaT}$ to represent MO to ensure that our simulation results are less dependent on the properties and mathematical formalism of a given single ion channel. Each simulation was color-coded by steady-state FR in BPM. For each CO-MO map, the border between no-firing and irregular firing regions (Figs 3 and 4; yellow lines) was identified by segregating zero and non-zero values of FR. Additionally, the border between irregular and rhythmic firing regions (Figs 3 and 4; blue lines) was established by pinpointing the location of the first local trough along the direction of CO and MO reduction.

To access SANC robustness, we quantify robustness as the percentage area of rhythmic firing regions in CO-MO parameter space maps (Figs 3 and 4). To evaluate SANC performance and flexibility, we quantify performance as the maximal time-of-day SANC FR in BPM, flexibility as the difference between the maximal and minimal time-of-day SANC FR in BPM with the control CO and MO. Sensitivity analysis was conducted with a full inhibition of each module of interest (Fig 5). Additionally, sensitivity analysis for $[Na]_i^+$ accumulation was performed by clamping time-of-day $[Na]_i^+$ content at its steady-state concentration at ZT6.

All model simulations were performed with parallel computing on a ThinkStation P620 tower workstation with an AMD Threadripper processor. Model codes were implemented and solved in MATLAB (Version: 9.13.0 (R2022b)) using ode15s, and will be provided for public download from https://github.com/Mathbiomed/CircSANC.

## Supporting information

**S1 Table. Model parameters and settings.**
(DOCX)

## Acknowledgments

The authors thank all members of the Biomedical Mathematics Group at the Institute for Basic Science for their most helpful discussions.

## Author Contributions

**Conceptualization:** Pan Li, Jae Kyoung Kim.

**Data curation:** Pan Li.

**Formal analysis:** Pan Li.

**Funding acquisition:** Jae Kyoung Kim.

**Investigation:** Pan Li.

**Methodology:** Pan Li.

**Project administration:** Jae Kyoung Kim.

**Resources:** Jae Kyoung Kim.

**Software:** Pan Li.

**Supervision:** Jae Kyoung Kim.

**Validation:** Pan Li.

**Visualization:** Pan Li, Jae Kyoung Kim.

**Writing – original draft:** Pan Li.

**Writing – review & editing:** Pan Li, Jae Kyoung Kim.

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
