## [Decision Letter · Decision Letter 0]

20 Oct 2023

Dear %TITLE% Kim,

Thank you very much for submitting your manuscript "Central and peripheral clocks synergistically enhance circadian robustness, flexibility and performance of cardiac pacemaking" for consideration at PLOS Computational Biology.

As with all papers reviewed by the journal, your manuscript was reviewed by members of the editorial board and by several independent reviewers. In light of the reviews (below this email), we would like to invite the resubmission of a significantly-revised version that takes into account the reviewers' comments.

The submitted manuscript utilized mathematical modeling to simulate cardiac pacemaking controlled by central and peripheral clocks uncovering potential impacts of aging in circadian regulation of heart rate. The reviewers acknowledge that synergistic interactions between PNA, SNA, and LCR regulating cardiac pacemaking are interesting. However, reviewers raised some major concerns including lack of technical details, potential effects of temperature, and impact of calcium signaling over aging. Importantly, it will be critical to clearly define the terms used in the manuscript improving clarity of conclusions and overall readability of the manuscript.

We cannot make any decision about publication until we have seen the revised manuscript and your response to the reviewers' comments. Your revised manuscript is also likely to be sent to reviewers for further evaluation.

Sincerely,

Christian I. Hong, Ph.D.

Academic Editor

PLOS Computational Biology

Daniel Beard

Section Editor

PLOS Computational Biology

The submitted manuscript utilized mathematical modeling to simulate cardiac pacemaking controlled by central and peripheral clocks uncovering potential impacts of aging in circadian regulation of heart rate. The reviewers acknowledge that synergistic interactions between PNA, SNA, and LCR regulating cardiac pacemaking are interesting. However, reviewers raised some major concerns including lack of technical details, potential effects of temperature, and impact of calcium signaling over aging. Importantly, it will be critical to clearly define the terms used in the manuscript improving clarity of conclusions and overall readability of the manuscript.

Reviewer's Responses to Questions

**Comments to the Authors:**

Reviewer #1: Summary: This report features a mathematical model describing autonomic control and LCR in sinoatrial nodal cells that attempts to capture distinct circadian patterns in adult and aged mice. Although the paper is interesting, a number of technical concerns require clarification, and additional modeling of how calcium oscillations change with aging is required.

Major Comments

1) The entire paper is about how “PNA, SNA, and LCR, … synergistically work together”. By definition, “synergistically” means that together they achieve more than in isolation. Please be specific and refer to your results for separate or concerted actions and clearly formulate (and prove) conclusions as claimed in your title. How much synergy (in quantitative terms) is achieved in your model? Please add this data in the abstract and provide clear conclusions after or within the discussion section.

2) In fig3C, I see only a synergistic effect for flexibility, while the title claims synergistic effects also for robustness and performance. Please explain.

3) In Methods you refer to three papers (27, 31, 40) for your model. You also say that “In this study, we present a novel mathematical model”. In any case, please be more specific and cite the closest oldest model published previously that was used as a basis for your novel model. Also, describe in full detail what is novel compared to previous models and, very importantly, please provide full computer code for your new model for all of your simulations. Your working code should be in the paper supplement or in GitHub or any other public database. This will allow anyone (incl. reviewers, editors, and readers) to reproduce your results and to use in further studies by running your actual (tested and working) code.

4) You wrote: “This modeling choice aligns with earlier experimental findings on the effects of a thermoneutral environment on heart rate variability, where mice exhibit a high sympathetic drive to maintain a normal core temperature (37°C) under standard laboratory conditions (20°C) (15, 36).” Please provide more details about how your model implemented the factor of temperature, and why and how specifically “This modeling choice aligns with earlier experimental findings”

5) “We modeled LCR as a diurnal variation in the expression level of IHCN, but it might not be the only mechanism underlying LCR in SANC.”

Yes, sure but this is a major study limitation of the study and should be discussed in more detail. You perform parametric sensitivity analysis for both CO and MO (i.e. Ca and membrane oscillators), but model LCRs only as one component in MO.

6) How were CO-MO parameter space maps created in Figs 3 and 4? You showed MO changes from 0 to 1 and CO also from 0 to 1. Which specific model parameters were varied, in what specific range, and why were these parameters and ranges chosen?

7) Explain pattern disruption (black band in the middle) in Fig.3 for ZT18 in SNAB and ANSB. There are also clear disruptions of BPM within rhythmic firing areas between the black area and blue line. This may indicate the way you changed model parameters in the sensitivity analysis for the MO and CO between 0 and 1, or possibly may indicate problems with your model integration or automatic analysis of the data. Please check and explain.

8) What algorithm was used to analyze the simulation data in Figs 3 and 4? In other terms, how did you get BPM in each simulation trace? This may make a big difference in the results, depending how you define AP, i.e. how you separate AP from a Vm oscillation near 0 mV.

9) In Figs 3 and 4 the black area indicates: “Chaotic or no firing”. There is a big difference between chaotic vs. no firing. Please separate “Chaotic” and “no firing”. Also, in many panels (e.g., Fig 3 ZT18-control) there is a mosaic pattern of BPM between the blue line and black area. Is this a mosaic pattern chaotic firing? If so, why then it is not within the black area that includes chaotic firing as per your description? It’s important to provide clear definitions for chaotic and rhythmic firing in your analysis.

10) What was your rational to perform sensitivity analysis with the key model parameter reduced exactly by 25% in Fig. 5B?

11) How did you define the specific model parameter changes in aging in Fig.5A? Was your choice based on experimental studies? If so, please cite the respective literature in the figure legend. If not, then explain your reasoning for the parameters you use in the simulations.

12) I found it very difficult to read the paper. It seems like cryptography with numerous abbreviations: FR, MO, CO, SNA, ANS, LCR, SANC, AP, NUC, PNA, PNA, SNAB, ANSB, HR, iHR, DCN, GPCR, PKA, AC, SR, HCN, SERCA. P-S, S-P, BPM. Further, many abbreviations are not explicitly given like ZT6 and ZT18. Please reduce abbreviations to a minimum to make the text readable. For example, why not write simply firing rate instead of FR or Ca oscillator instead of CO, etc.? Also please include a table defining all terms.

13) Ist has a strong contribution to your model, whereas molecular identity of this current remains unknown. The same is about INab, and Cab. Please explain.

14) In Fig.1 only 3 currents are affected by aging (ICaL, ICaT, I HCN), all other 13 current remain unchanged, including currents of major functional importance, such as NCX, NaK, KACh, Ito, Na currents. Please explain why only three currents change with aging in your model. What is known about how the other 13 currents change with aging. If some or all these currents do change with aging, your results would likely be different?

15) In Fig.1 Jup and Jrel (and the entire Ca oscillator module - CO) are not affected by aging. This is not correct. There is evidence that Ca signaling (SERCA, RyR, NCX) deteriorates with age, see e.g. Liu et al. Am J Physiol Heart Circ Physiol. 2014;306(10):H1385-97. Thus, your model does not reproduce a major change in the Ca oscillator. Please explain or make respective additional simulations.

///////////////////////////////////////////////////////////

Minor Comments

1) Please clearly define Robustness, Flexibility, and Performance, and explain why those definitions are reasonable.

2) You wrote: “This AC cAMP-PKA signaling cascade leads to the phosphorylation of various targets, such as the L-type Ca2+channel (ICaL), Ryanodine receptor, and sarco/endoplasmic reticulum (SR) Ca2+ ATPase (SERCA)”

In fact, PKA activates SERCA by phosphorylation of phospholamban, is there any evidence that SERCA, itself becomes phosphorylated?

Reviewer #2: The authors develop a mathematical model designed to predict the impact and importance of various factors (parasympathetic and sympathetic nervous system and intrinsic gene changes) in circadian regulation of heart rate (HR). The authors conclude that “our work shed light on their critical synergistic interactions in regulating time-of-day cardiac pacemaking function and dysfunction, which may help to identify potential therapeutic targets within the circadian clock for the prevention and treatment of cardiac arrhythmias.”

Reviewer #3: “Central and peripheral clocks synergistically enhance circadian robustness, flexibility

and performance of cardiac pacemaking” by Li and Kim is an ambitious attempt to understand how 24-hour oscillations in receptor-mediated signaling and ion channel expression impact sinoatrial nodal cells' firing rate and functionality. They extend the modelization to include a model of sinoatrial node cells in aged mice. The authors are commended for their approach to model a complex interaction at a new level of detail. My high enthusiasm for the work is somewhat diminished by the lack of data showing the temperature's impact on the system. Temperature addition may be complex, but a fundamental component of intrinsic circadian signaling is the daily rhythm in core body temperature. The addition is expected to impact the results of the work and perhaps its conclusions. This limitation should be addressed. Another challenge is that the article contains a lot of jargon and abbreviations. The jargon and the numerous non-standard abbreviations lower the understandability and readability of the article. The authors are encouraged to define terms clearly. The authors are encouraged not to use many non-standard abbreviations to improve readability. Lastly, some aspects of the premise appear misrepresented or overstated.

Concerns:

Please provide a clear, succinct definition for the following terms in a table: circadian rhythms (this has many different meanings based on the field), circadian flexibility, reliance and adaptability of circadian rhythms, and circadian robustness. The authors are encouraged to use terms/definitions widely accepted by chronobiologists. This should broaden research interest in this important work.

Please reduce the number of non-standard abbreviations to improve readability.

Please discuss or include temperature rhythms. A hallmark feature of biological rhythms is that the 24-hour rhythm in core body temperature serves as an intrinsic Zeitgeber for local circadian clocks in the peripheral tissues. Please discuss how a daily rhythm in core body temperature may play a role in the modelization. Can the authors include this concept in the simulations?

Figure 2A data is based on data from Barazi. The authors show hourly data with error bars, but the original article shows around five-time points. How was the data in Figure 2A generated? The authors are encouraged to show only the actual data or alter the description of Figure 2A. It is misleading as presented.

The Barazi article also suggests no significant fluctuation in the 24-hour rhythm in the heart rate with ANS block or PNS block (Figure 1 in Barazi). Why do the authors discuss these data sets as being circadian? What was the data that the authors used to make this conclusion?

ANS block data in Figures 2A and 2B are qualitatively different when the amplitude peaks (ZT 24 vs ZT 18). Why?

Reviewer #4: This paper is focused on circadian regulation of cardiac excitability, an important aspect of physiology that has not been thoroughly addressed in previous computational modeling studies. The authors extend a published model of action potential generation in mouse sinoatrial node cells (SANC) to incorporate the regulation of cardiac pacemaking by both the central circadian clock—through the autonomic nervous system--and local circadian rhythmicity (LCR) within the heart. The authors perform simulations and sensitivity analysis to assess the role of parasympathetic nervous activity (PNA), sympathetic nervous activity (SNA), and LCR on circadian rhythms in heart rate (HR). They also use their model to study mechanisms underlying the age-related decline of circadian rhythms in HR.

Overall, the study is well-motivated, and the manuscript is well-written. The findings on the importance of non-additive interactions between PNA and SNA for reproducing circadian patterns in SANC firing rate are potentially interesting. My main concern is that the evidence supporting this finding is not yet adequately explained in the manuscript, as I describe in my comments below.

Major comments

1) It is stated on lines 175-178 that when additive PNA and SNA (no PNA-SNA interactions) is assumed, the baseline of the simulated circadian rhythms is approximately 5% lower than the experimental data. I don’t understand how this 5% is calculated… for example, it looks like the trough of the simulated curve is 400 BPM at ZT6 (Fig 2B, grey boxes) compared to a trough of 500 BPM in the experimental data at ZT6 (Fig 2A, grey dots), which would correspond to a 20% difference. A similar calculation using the peak of the simulated and experimental curves at ZT18 (475 and 575 BPM, respectively) gives a 21% difference. How was the 5% result obtained?

2) The authors go on to say that when they implemented bidirectional modulation effects, the circadian pattern in FR under the control condition was accurately simulated (Fig 2B, grey line). The trough and peak of this simulated curve appear to be at 425 and 500 BPM at ZT6 and ZT18, which are still 17% and 15% lower than the corresponding experimental values, respectively. Thus, more explanation is needed to justify the claim of accurate simulation as well as the statement on line 199-200 that the incorporation of bidirectional effects “aligns the simulated FR with experimental data (A; grey dots)”.

3) Based on the above calculations, the simulated non-additive curve (grey line) does indeed agree better with the experimental curve (grey dots) than the simulated additive curve (grey boxes) by a few percent, but given the large error bars associated with the grey dots in the experimental data (Fig 2A) it is not clear to me that there is enough evidence to support the strong claim on lines 186-187 that the “results unequivocally indicate that the non-additivity of PNA and SNA is required to properly construct the circadian patterns of SANC function”.

Is it not possible that with other parameter choices, the experimental data could be equally well-recapitulated with purely additive effects?

4) The authors write on lines 183-184 that both minimal and maximal time-of-day FR are in quantitative agreement with the experimental data (Fig 2C). The caption of Fig 2C explains that the FR values were normalized to FR at ZT12 under the ANSB condition. I’m afraid I don’t quite understand how this normalization was applied. The curve of green dots in Fig 2A appears to reach a minimum at ZT12, thus I would expect the Normalized HR Min value under the ANSB condition to be exactly 1 for the experimental data. However, in Fig 2C the Normalized HR Min value under the ANSB condition shown for the experimental data is less than 1.

5) Why does the simulated FR curve under ANSB condition (Fig 2B green line) have a trough at ZT6, whereas the experimental FR curve under ANSB condition (Fig 2A green dots) has a trough at ZT12?

Does this discrepancy affect the appropriateness of using FR at ZT12 under ANSB condition as the reference value for normalization?

6) In general, why are the simulated FR lower than the experimental HR, as evidenced by the lower range used for the y-axis in Fig 2B (300 to 550 BPM) than Fig 2A (350 to 700 BPM)?

It is not necessary for a model to exactly reproduce the same range of values as seen in experiments to be useful, but since the simulated FRs are systematically and significantly lower than the experimental HRs it would be helpful for the authors to explain or at least comment on this discrepancy. Furthermore, this difference between the simulated FR and the experimental HR suggests that it might be more appropriate to describe the results as “qualitative” rather than “quantitative” reconstructions of diverse circadian patterns in SANC FR the section heading on line 164 and the Fig 2 caption. Similarly, is it accurate to say on line 173 that the model simulations “meticulously recapitulated” the experimental findings given that there is about a 100 BPM difference between the two?

7) More details on how the model was calibrated to experimental data should be provided. For example, on lines 55-57 the authors say the bidirectional modulation effects P-S and S-P were formulated and calibrated to account for the non-additivity effects (8-11) but it is not explained what these effects are.

Similarly, it is mentioned that the model was calibrated for time-of-day PNA (line 420), time-of-day SNA (line 423), age-dependent MO reduction (line 436), aging-dependent PNA impairment (line 439), but it is not described how exactly the model was calibrated. Were the data sufficient to constrain the parameters, or are there multiple parameter sets that can reproduce the data well?

Some discussion of parameter identifiability and how the authors modeling approach and parameter fitting does or does not account for the variability/heterogeneity of cardiac electrophysiology in general (and more specifically, the variability/heterogeneity depicted by the error bars in Fig 2A) seems warranted.

8) Is there a way to experimentally test the model prediction that non-additive interactions between PNA and SNA are important for the circadian patterns, or to experimentally validate that these interactions were modeled appropriately? If so, it would be helpful to discuss these possibilities in the Discussion section.

Minor Comments

1) The authors conclude on lines 156-158 that SNA serves as a SANC robustness and performance enhancer, PNA acts as a flexibility amplifier, and LCR functions as a flexibility and performance booster.

Do the words “enhancer, amplifier, and booster” as used here all mean the same thing, or were the 3 different words employed to indicate some differences in the roles that SNA, PNA, and LCR play when it comes to interpreting their simulation results? For example, would it be just as accurate to say that PNA acts as a flexibility booster or enhancer instead of as a flexibility amplifier? If so, then the authors should just pick one of these 3 words to use throughout this sentence (as well as in lines 346-349) so that different meanings aren’t implied.

2) The phrase “SNA in this study was modeled as a high ISO tone” on lines 52-52 struck me as strange as I do not usually see the word tone associated with a drug. Also, a brief explanation of why using isoproterenol to model SNA would be helpful even if it was previously described in (3).

3) In line 144 of the Abstract, replace “Our work shed light” with “Our work sheds light”

4) An easier-to-follow explanation of what simulations were performed to obtain the curves in Fig 2B would be helpful for readers to be able to reproduce the results. All of the information may already be provided in the Materials and Methods or Supporting Information, but it is not explicitly spelled out. For example, to go from the Control curve to the P-S curve, does one simply set the parameter k_{s_p} to 1? Similarly, what parameters would one change to go from the Control curve to the S-P curve, the Additive curve, or the 3 blockade curves? If the authors would prefer not to include that type of description in the manuscript itself, it could go in the readme file associated with the model code instead.

**Have the authors made all data and (if applicable) computational code underlying the findings in their manuscript fully available?**

Reviewer #1: **No: **See comments to authors

Reviewer #2: **No: **

Reviewer #3: **No: **They will after acceptance

Reviewer #4: Yes

PLOS authors have the option to publish the peer review history of their article (what does this mean?). If published, this will include your full peer review and any attached files.

Reviewer #1: **Yes: **Edward G. Lakatta, MD

Reviewer #2: No

Reviewer #3: No

Reviewer #4: No
---

## [Decision Letter · Decision Letter 1]

12 Feb 2024

Dear Prof Kim,

We are pleased to inform you that your manuscript 'Circadian regulation of sinoatrial nodal cell pacemaking function: dissecting the roles of autonomic control, body temperature, and local circadian rhythmicity' has been provisionally accepted for publication in PLOS Computational Biology.

Best regards,

Christian I. Hong, Ph.D.

Academic Editor

PLOS Computational Biology

Daniel Beard

Section Editor

PLOS Computational Biology

The revised manuscript addressed all of the previous concerns raised by the reviewers. One of the reviewers insists that the authors upload their model code to a public repository. Please make sure to deposit your model into an open software archive following the PLoS Computational Biology guidelines.

Reviewer's Responses to Questions

**Comments to the Authors:**

Reviewer #1: The author’s have adequately responded to my comments and have satisfactorily modified the revised manuscript. The response to my comment #3 is not satisfactory.

My comment #3 …Your working code should be in the paper supplement or in GitHub or any other public database. This will allow anyone (incl. reviewers, editors, and readers) to reproduce your results and to use in further studies by running your actual (tested and working) code.

Your response was ‘..will upload model codes underlying our study to our group’s website on GitHub for public download once the manuscript is accepted’

Reviewer #2: My comments have adequately addressed.

Reviewer #3: The authors have addressed my concerns. The inclusion of temperature has elevated the impact of the paper.

Reviewer #4: The authors have thoroughly revised the manuscript and addressed my concerns.

**Have the authors made all data and (if applicable) computational code underlying the findings in their manuscript fully available?**

Reviewer #1: **No: **See Comments to Editor

Reviewer #2: Yes

Reviewer #3: Yes

Reviewer #4: Yes

PLOS authors have the option to publish the peer review history of their article (what does this mean?). If published, this will include your full peer review and any attached files.

Reviewer #1: **Yes: **Edward G. Lakatta, MD

Reviewer #2: No

Reviewer #3: No

Reviewer #4: No

---

## [Editor Report · Acceptance letter]

21 Feb 2024

PCOMPBIOL-D-23-01268R1 

Circadian regulation of sinoatrial nodal cell pacemaking function: dissecting the roles of autonomic control, body temperature, and local circadian rhythmicity

Dear Dr Kim,

I am pleased to inform you that your manuscript has been formally accepted for publication in PLOS Computational Biology. Your manuscript is now with our production department and you will be notified of the publication date in due course.

With kind regards,

Bernadett Koltai
